# CityRefer: Geography-aware 3D Visual Grounding Dataset on City-scale Point Cloud Data

**Taiki Miyanishi**[1,3*], **Fumiya Kitamori**[2*], **Shuhei Kurita**[3],
**Jungdae Lee**[2], **Motoaki Kawanabe**[1], **Nakamasa Inoue**[2]

[1]ATR,   [2]Tokyo Institute of Technology,   [3]RIKEN AIP

## Abstract

City-scale 3D point cloud is a promising way to express detailed and complicated outdoor structures. It encompasses both the appearance and geometry features of segmented city components, including cars, streets, and buildings, that can be utilized for attractive applications such as user-interactive navigation of autonomous vehicles and drones. However, compared to the extensive text annotations available for images and indoor scenes, the scarcity of text annotations for outdoor scenes poses a significant challenge for achieving these applications. To tackle this problem, we introduce the *CityRefer dataset*[1] for city-level visual grounding. The dataset consists of 35k natural language descriptions of 3D objects appearing in SensatUrban [19] city scenes and 5k landmarks labels synchronizing with OpenStreetMap. To ensure the quality and accuracy of the dataset, all descriptions and labels in the CityRefer dataset are manually verified. We also have developed a baseline system that can learn encoded language descriptions, 3D object instances, and geographical information about the city's landmarks to perform visual grounding on the CityRefer dataset. To the best of our knowledge, the CityRefer dataset is the largest city-level visual grounding dataset for localizing specific 3D objects.

## 1 Introduction

Advancements in urban 3D scanning technologies, such as unmanned aerial vehicle photogrammetry and mobile laser scanning, enable the creation of accurate and photorealistic large-scale 3D scene datasets. Examples of such datasets include street-level datasets acquired by automobiles [6, 17, 37, 39, 42, 43] and city-level datasets acquired by aerial vehicles [19, 25, 36, 44, 52, 60]. However, while city-level photorealistic 3D scans have become practical and applicable in various fields like autonomous driving and unmanned vehicle delivery, the technology to comprehend city scenes through human-interactive linguistic representations is still in its early stages of development. The ability to ground linguistic expressions to urban components is highly desired for interactive and interpretable applications, such as language-guided autonomous driving and aerial drone navigation. Achieving this requires the development of a 3D visual grounding dataset based on the city-scale point clouds, which presents a significant challenge.

3D Visual grounding is a 3D and language task that involves localizing objects in 3D scenes based on textual referred expressions. Compared to its 2D visual grounding counterparts [23, 29, 31, 53], 3D visual grounding poses additional challenges. The expressions used in 3D visual grounding often require more information to localize object instances due to the rich context of 3D scenes, making the problem further complex. Recent studies have made remarkable progress in 3D visual

---

*equal contribution

[1]https://github.com/ATR-DBI/CityRefer

37th Conference on Neural Information Processing Systems (NeurIPS 2023) Track on Datasets and Benchmarks.

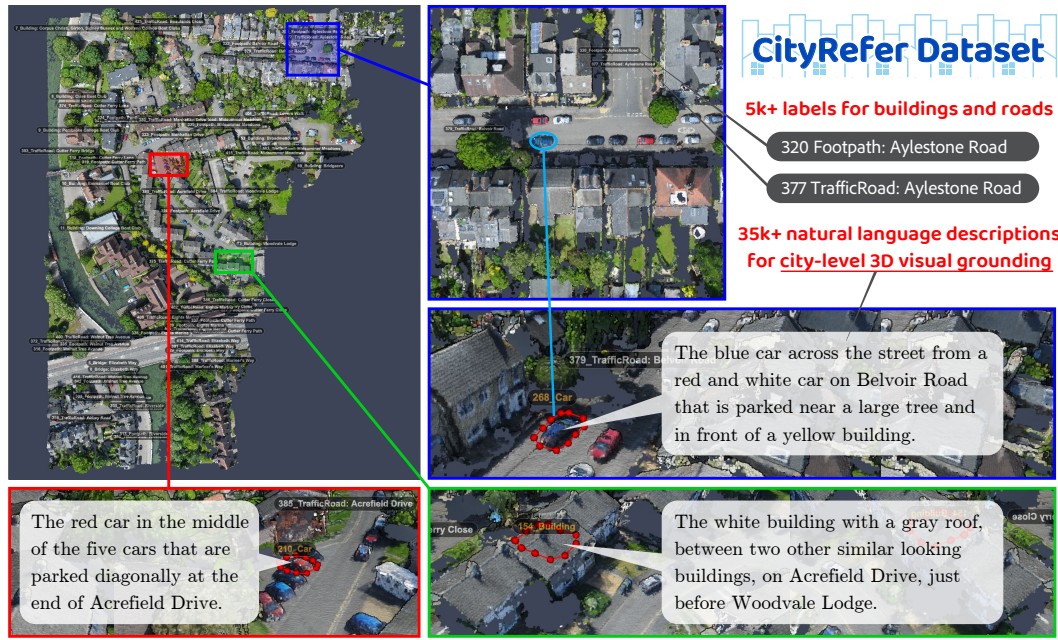

Figure 1: **The CityRefer dataset for city-level 3D visual grounding.**

grounding, focusing on determining the precise position of 3D objects given natural language descriptions [7, 10, 18, 20, 21, 22, 35, 54, 56, 58]. However, many of these methods have been evaluated on 3D indoor datasets, which typically consist of point clouds of room scenes and labels for household objects [2, 9, 14]. Subsequently, Kolmet *et al.* [24] proposed a district-based visual grounding dataset on KITTI-360 [26]. Nevertheless, the availability of 3D visual grounding datasets is still limited, especially in the context of aerial city-level 3D point clouds. Therefore, we aim to address this gap by creating a publicly available 3D visual grounding dataset based on aerial city-level 3D point clouds.

In this paper, we introduce the *CityRefer dataset* for city-level 3D visual grounding. Specifically, we provide 35k natural language descriptions to localize 3D objects in the SensatUrban [19] environment as well as 5k labels of objects such as buildings and roads. Three example descriptions of a sample scene are shown in Figure 1. As seen from the figure, city-level visual grounding is very challenging because a system is required to find objects from a wide city area while understanding the description of the target object and the relationships between relevant objects. Although we used crowdsourcing to scale up the annotation, we needed to thoroughly filter out incorrect annotations by hand to finalize the dataset and ensure the quality of the annotations. The main contributions of the CityRefer dataset are summarized as follows.

1. We provide instance-wise segmentation masks for 5k objects including 1.8k landmark objects with their name labels. Examples include *Kem River and Baker Street, Aylestone Road, and Belvoir Road*. These labels were obtained from the spatial joint between SensatUrban and OpenStreetMap using our semi-automatic system (Section 3.1).

2. We provide 35k natural language descriptions for city-level visual grounding. These descriptions are manually attached using our interactive annotation system (Section 3.2).

3. We provide a baseline system that performs city-level 3D visual grounding. Because it is nontrivial to adapt previous visual grounding methods for our city-level dataset, we propose a simple but effective method that narrows the search area to find the target object by using geographical information.

Table 1: Comparison of 3D visual grounding datasets. $N_{\mathrm{desc}}$ : Number of natural language descriptions. $\bar{L}_{\mathrm{desc}}$ : Average description length. $N_{\mathrm{points}}$ : Number of 3D points.

| | Dataset | Human annot. | $N_{\mathrm{desc}}$ | $\bar{L}_{\mathrm{desc}}$ | Area | Environment (Format) | $N_{\mathrm{points}}$ |
|---|---|---|---|---|---|---|---|
| Indoor | REVERIE [32] | Yes | 21,702 | 18.0 | Rooms | Matterport3D (RGB) [8] | - |
| | SUN-Spot [30] | Yes | 7,987 | 14.1 | Rooms | SUN RGB-D [40] | - |
| | SUNRefer [27] | Yes | 38,495 | 14.1 | Rooms | SUN RGB-D [40] | - |
| | Nr3D [2] | Yes | 41,503 | 11.4 | Rooms | ScanNet (3D Scan) [14] | 242M |
| | ScanRefer [9] | Yes | 51,583 | 20.3 | Rooms | ScanNet (3D Scan) [14] | 242M |
| Outdoor | TouchDown [11] | Yes | 25,575 | 29.7 | Roadside | Google Street View (RGB) | - |
| | KITTI360Pose [24] | No | 43,381 | 7.6 | Roadside | KITTI-360 (3D Scan) [26] | 1,000M |
| | CityRefer (Ours) | Yes | 35,196 | 26.3 | City center | SensatUrban (3D Scan) [19] | 2,847M |

## 2 Related Work

**Visual Grounding Datasets for 3D Spaces.** Over the years, there has been significant research interest in 3D visual grounding as summarized in Table 1. We first introduce two types of 3D visual grounding datasets, each focusing on a different level of the scene: the indoor and outdoor roadside. The 3D visual grounding dataset is created by annotating indoor or outdoor 3D datasets with linguistic descriptions.

**(i) Indoor scene level.** With the increasing availability of indoor 3D datasets [3, 5, 8, 14, 46, 34, 40, 49], several visual grounding datasets have been proposed to address the demand for 3D scene understanding. REVERIE [32] comprises 10,318 panorama images captured across 86 indoor scenes and a total of 4,140 target objects. This dataset also provides 21,702 language instructions with rich textual annotations for guiding virtual agents within indoor photorealistic scenes of Matterport3D [8]. The SUN-Spot [30] and SUNRefer [27] datasets focus on object localization in single-view RGB-D images within indoor environments from the SUN RGB-D dataset [40]. Both datasets provide detailed language annotations indicating the spatial extent and location of objects in the images including object bounding boxes. Specifically, SUNRefer contains 38,495 language annotations for 7,699 objects in indoor RGB-D images. Nr3D [2] and ScanRefer [9] are standard 3D visual grounding datasets that are built on top of ScanNet [14], a real-world 3D scene dataset with extensive semantic annotations. However, these datasets mainly focus on the indoor visual grounding task.

**(ii) Outdoor scene level.** In recent years, a number of richly annotated outdoor 3D datasets have been created by scanning cities using sensors installed in cars and drones [6, 17, 19, 25, 37, 42, 44, 50, 60]. While there have been efforts to annotate outdoor 3D datasets with language descriptions for 3D visual grounding, the availability of such datasets is still limited compared to indoor ones. The TouchDown dataset [11] is aimed at text-guided navigation and spatial reasoning using real-life visual observations. It contains 9,326 examples of instructions and spatial descriptions in the visual navigation environment drawn from Google Street View. KITTI360Pose [24] is a text-based position localization dataset in an outdoor 3D environment from the KITTI360 dataset [50], which provides nine static scenes obtained by LiDAR scans. It is notable that the linguistic descriptions of KITTI360Pose are automatically generated by a sentence template with position description query pairs. While both TouchDown and KITTI360 datasets are based on the vehicle perspective and hence limited to the semantics from roadsides, our dataset is based on SensatUrban [19], which covers 3D semantics of the board city areas that are generated from aerial images by drones.

**Learning Visual Grounding of 3D Scenes.** To facilitate a deeper understanding of 3D scenes through language, there have been many efforts to connect languages and the 3D visual world, including 3D dense captioning [12, 47, 55], 3D change detection [33], and 3D visual question answering [1, 4, 15, 28, 59]. Specifically, 3D visual grounding that aims to locate an object in 3D space in response to a natural language query is the fundamental task of the 3D and language field [2, 9, 48]. Several approaches in the field of visual grounding use pre-computed 3D object detection or instance segmentation results, utilizing the point cloud features extracted from the corresponding 3D bounding boxes and segments [20, 56]. However, the challenge of object recognition arises due to the low resolution of 3D data resulting from the reconstruction process. To overcome this limitation, recent

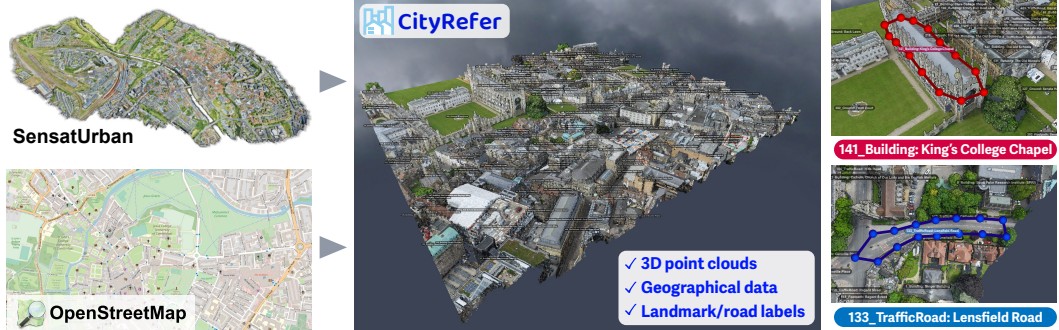

Figure 2: **Stage 1 of dataset construction.** We perform a spatial join between the 3D SensatUrban environment and the 2D OpenStreetMap. The CityRefer dataset contains 1,850 landmark/road labels and geographical data. Examples include the landmark name *King's College Chapel* and the road name *Lansfield Road*.

studies have proposed the integration of both 2D images and 3D data [51, 22]. By combining the rich spatial information provided by 3D data with the detailed appearance cues derived from 2D images, these hybrid approaches aim to enhance robustness in the context of visual grounding. Furthermore, there have been studies proposing methods that integrate 3D visual captioning and grounding, where both models are learned simultaneously to achieve synergistic effects [7, 10]. These current 3D visual grounding methods mainly rely on two widely used indoor 3D visual grounding datasets [2, 9]. Several studies proposed visual grounding on remote sensing data but were limited to 2D images [41, 57]. For these reasons, the performance of 3D visual grounding on outdoor 3D datasets remains unexplored. One of the initial studies [24] attempted to identify regions within the 3D point cloud based on textual queries on city-level 3D datasets but were limited to artificial language descriptions and did not fully use the geographic information 3D map despite the sparsity of city-level 3D point cloud data. In contrast, our method uses geographic information from 3D maps to achieve accurate city-level 3D visual grounding.

## 3 Dataset Construction

The CityRefer dataset consists of 1) instance-wise segmentation masks each with a label and geographical information, and 2) natural language descriptions for visual grounding. The 3D environment we use is the SensatUrban [19], which consists of photogrammetric point clouds of two UK cities covering 6 km$^2$ of the city landscape. Semantic segmentation masks are provided with the environment, and our study refines them to instance-level masks for visual grounding. The annotation proceeds in two stages: semi-automatic generation of instance-wise segmentation masks (Stage 1) and manual language annotation (Stage 2).

### 3.1 Stage 1: Semi-automatic Generation of Instance-wise Segmentation Masks

The goal of this stage is to generate segmentation masks for each 3D object as well as to attach labels of names with geographical data (longitude, latitude, and elevation) to each object. To achieve this, we perform a spatial join between the 3D environment and OpenStreetMap, from which we can obtain names and locations. Figure 2 shows an example result of this stage. As shown, regions of *King's College Chapel* and *Lensfield Road* are visualized precisely. The procedures are described in detail below.

**Georeferencing.** The georeferencing is performed in the following three steps. First, given a block[2] of the 3D city, we create a top view image as shown in Figure 3a, where the image size is fixed to 2048 by 2048 pixels. Second, we manually choose ten points of interest, such as corner and landmark points, that are clearly visible in OpenStreetMap. In this step, we also extract an image of the 2D map as shown in Figure 3c. Note that the coordinates of the ten points are manually annotated on

---

[2]We use 34 blocks (scenes) of Cambridge and Birmingham from SensatUrban.

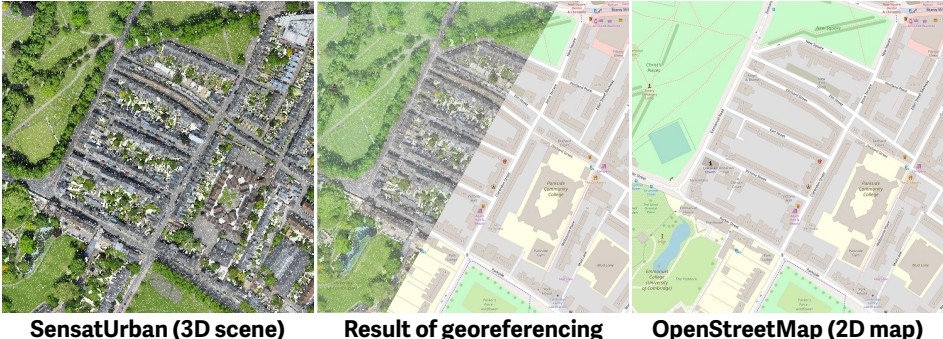

| SensatUrban (3D scene) | Result of georeferencing | OpenStreetMap (2D map) |

Figure 3: Georeferencing of 3D environment and 2D map.

both images. Finally, we compute the geometric transformation between the two images by using the transformation function from the scikit-image library. We manually tune the hyper-parameters of transformation by visually verifying the results of the georeferencing. Here, the ten points of interest are also updated and tuned if needed. Figure 3b shows an example result in which the 3D scene and the 2D map are precisely joined.

**Generating instance-wise segmentation masks.** Given semantic segmentation masks with respect to 13 object categories, we refine them to instance-level masks. We divide the categories into three groups as shown in Table 2. For Group 1, we directly use geographic information obtained from OpenStreetMap to create filters for each instance. For example, for a segment of *Lansfield Road* in OpenStreetMap, we create a filter in the 3D scene based on the result of the georeferencing.

For Group 2, we apply the DBSCAN algorithm, a clustering method, to 3D points. It is not difficult to obtain accurate boundaries between instances by clustering because objects in this group are small and located separately. For Group 3, we used YOLOv7 to detect cars. The detection results are not perfect but are sufficient for creating the dataset. Note that we ask annotators to only use correctly segmented instances in the next stage.

Table 2: Three groups of categories.

| Grp. | Method | Categories |
|---|---|---|
| 1 | Filtering | Ground, HighVegetation, Building, Bridge, Rail, TrafficRoad, Footpath, Water |
| 2 | Clustering | Wall, Parking, StreetFurniture, Bike |
| 3 | Detection | Car |

## 3.2 Stage 2: Manual Language Annotation

The goal of this stage is to collect natural language descriptions that describe the target object in a 3D scene for visual grounding. We prepare two interfaces: the language annotation interface and the quality control interface. The former is used to collect descriptions, and the latter is used to verify whether the collected descriptions are accurate. Below we present details of the interface implementation, the language annotation task, and the quality control procedure.

**Interface.** Figure 4 shows the interface we used for language annotation and verification. To interactively show 3D scenes to annotators, we implement the interface with Potree [38], an open-source WebGL-based point cloud renderer that can process large-scale point cloud data. In the figure, the target object is highlighted in red. We provide interactive features such as zooming and panning as well as the labels of each object. The annotators can view the regions of each object by clicking the labels.

**Language annotation.** We ask annotators to describe the target object with the following instructions.

> You will see geographic objects in different 3D outdoor scenes. Please describe the objects in the 3D scene **so that the objects can be uniquely identified** on the basis of your descriptions and what you observed when submitting your responses. Some of the information below (a combination is preferred) should be included in your description:
>
> - The object's appearance, *e.g., colors, shapes, materials.*
> - The object's location in the scene, *e.g., the parking lot is in front of Birmingham Library.*

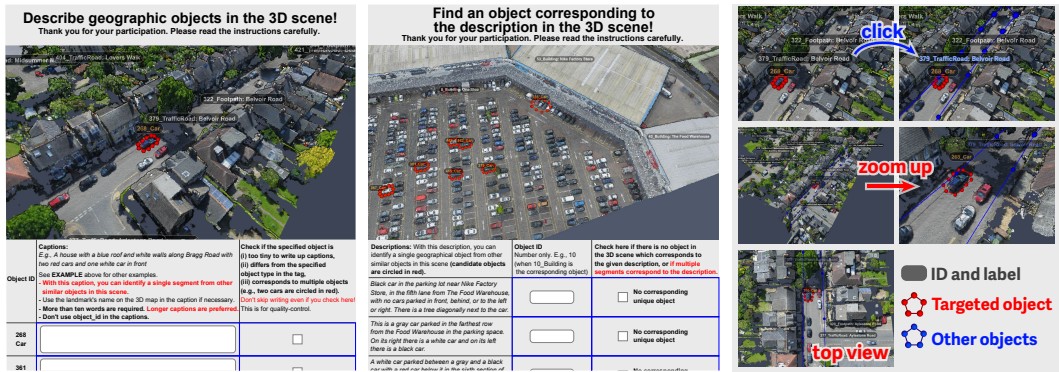

Figure 4: **Stage 2 of dataset construction.** We ask annotators to describe the target object in the 3D scene so that the object can be uniquely identified based on the description. With our interface, annotators can see each object's region. Functions such as zooming and panning are also available.

> - The spatial relation between this object and other objects, *e.g., this building is the second one from the left.*
>
> Imagine you and your friend live in a certain city, and you would like to ask your friend to find a geographic object in that city. Since there may be many similar geographical objects, the description of the object should be as unique as possible.

To efficiently collect data, we show at most three target objects to annotators in a 3D scene. We also ask annotators to report segmentation errors and incorrect labels via another free-form text box and checkboxes.

**Quality control.** To ensure the quality of the annotations, we ask another set of annotators to manually perform visual grounding with the following instructions.

> You will see 1-3 descriptions for different geographic objects. Please choose the geographical object that best matches the description from the 3D scene (candidate objects are circled in red.)

Based on the results, we remove incorrect descriptions. We measured the accuracy of the annotation with visual grounding again, and the correct response rate was 91.53%.

We used Amazon Mechanical Turk (MTurk) for annotation and quality control. There were 918 hours of work with a total cost of $9,699 (the estimated hourly rate paid was $10.56). The total number of participating workers was 282.

## 4 Dataset Statistics

This section provides basic statistics of the CityRefer dataset in comparison to ScanRefer [9] (an indoor dataset) and KITTI360Pose [24] (a roadside dataset). We summarize the statistics in Table 3 and discuss them in detail below.

**Target objects and descriptions.** The CityRefer dataset consists of 35,196 descriptions, each of which describes an object in the 3D scenes. The target objects fall under one of four categories: Car, Building, Ground, and Parking. The distribution is shown in Figure 5 (left). Note that all of them are unnamed objects; that means that landmark objects such as famous buildings and roads, which can be identified by their names on OpenStreetMap, are excluded from the target objects. The distribution of description lengths is presented in Figure 6, along with those of ScanRefer and KITTI360Pose for comparison. Here, KITTI360Pose exhibits a sharp length distribution of the descriptions because they are automatically generated from a database with templates, e.g., *the pose is south of a gray road*. In contrast, our dataset provides moderate-length descriptions to perform visual grounding. To the best of our knowledge, the CityRefer dataset is the first large-scale dataset with manually annotated descriptions of city-level 3D scenes.

Table 3: **Comparison of datasets.** Area: Type of scanned area. Manual: Whether annotation is manual or not. Geo data: Availability of geographical data. $N_{\text{desc}}$: Number of descriptions. $N_{\text{obj}}$: Number of objects. $N_{\text{landmark}}$: Number of landmark objects. $V$: Vocabulary size.

| Dataset | Area | Manual | Geo data | $N_{\text{desc}}$ | $N_{\text{obj}}$ | $N_{\text{landmark}}$ | $V$ |
|---------|------|--------|----------|-----------|----------|---------------|-----|
| ScanRefer [9] | Indoor | ✓ | | 51,583 | 11,046 | 0 | 4,197 |
| KITTI360Pose [24] | Roadside | | | 43,381 | 6,800 | 0 | 41 |
| CityRefer (Ours) | City center | ✓ | ✓ | 35,196 | 5,866 | 1,850 | 6,683 |

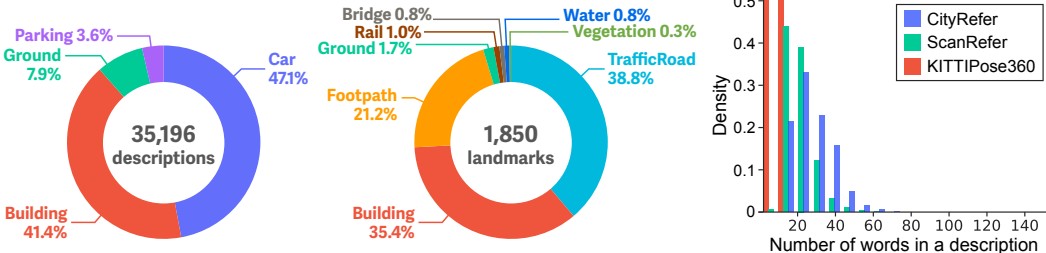

Figure 5: Object category distribution.    Figure 6: Description lengths

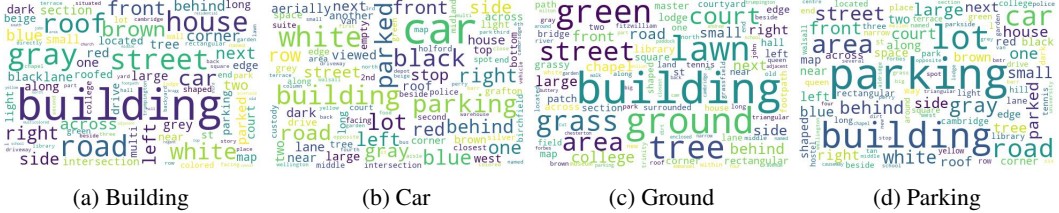

(a) Building    (b) Car    (c) Ground    (d) Parking

Figure 7: Visualization of word distributions.

**Landmark objects.** There are 1,850 landmark objects with their names across seven categories: TrafficRoad, Building, Footpath, Ground, Rail, Bridge, Water, and Vegetation. The distribution is shown in Figure 5 (right). Examples include *Senate House Hill*, *Wellhead Lane*, and *Parkside Police Station*. They are used to refer to target objects, *e.g., the gray rectangular building to the right of the parking lot next to St. John's College Chapel*.

**Words.** The vocabulary size of the CityRefer dataset is 6,683. Figure 7 shows the visualization of word distributions. We can observe that words that specify object locations in the following three categories are frequently used: 1) colors *e.g., red, blue, gray*, 2) relative positions *e.g., right, left, across*, and 3) nearby objects *tree, street, road*. Compared with the template-based position descriptions of KITTI360Pose, our free-form descriptions contain more natural descriptions.

## 5 Experiments

Finally, we conducted experiments on the CityRefer dataset, focusing on two tasks: instance segmentation and visual grounding. To ensure comprehensive evaluation, we divided the dataset into three subsets: training, validation, and testing. The data split is summarized in Table 4, providing the number of descriptions ($N_{\text{desc}}$), objects ($N_{\text{obj}}$), and landmark objects ($N_{\text{lmark}}$).

### 5.1 City-level instance segmentation

This experiment focuses on evaluating the segmentation performance. The objective is to generate segmentation masks with corresponding category labels for each object using 3D point clouds as input.

Table 4: Dataset split for training, validation, and testing

| Subset | $N_{\mathrm{desc}}$ | $N_{\mathrm{obj}}$ | $N_{\mathrm{lmark}}$ |
|---|---|---|---|
| Train | 23,586 | 3,931 | 1,106 |
| Val | 5,934 | 989 | 243 |
| Test | 5,676 | 946 | 501 |
| Total | 35,196 | 5,866 | 1,850 |

Table 5: Instance segmentation performance.

| Target | AP | $AP_{50}$ | $AP_{25}$ | mRec | $mRec_{50}$ | $mRec_{25}$ |
|---|---|---|---|---|---|---|
| Ground | 19.9 | 39.8 | 52.9 | 28.0 | 47.2 | 60.6 |
| Building | 3.7 | 12.2 | 24.4 | 7.2 | 16.8 | 26.5 |
| Parking | 5.9 | 17.0 | 48.3 | 15.1 | 30.8 | 59.0 |
| Car | 35.3 | 55.0 | 69.4 | 42.2 | 58.9 | 70.9 |
| Average | 16.2 | 31.0 | 48.7 | 23.1 | 38.4 | 54.2 |

**Method.** We provide a PyTorch-based implementation of a baseline method using the SoftGroup++ model [45]. Due to the larger number of 3D points (2,847M points) compared to previous datasets for 3D instance segmentation, we randomly sample 2% of the 3D points and feed them into the model. To provide transparency and reproducibility, we summarize the training details along with manually tuned hyperparameters in the Appendix.

**Evaluation metrics.** We use average precision (AP) and mean recall (mRec) as our primary metrics. In addition, we report the top-$N$ AP and recall, with $N$ set to 50 and 25. Specifically, we refer to these metrics as $AP_{50}$, $AP_{25}$, $mRec_{50}$, and $mRec_{25}$.

**Results.** Table 5 shows the resulting performance for the four target categories. We see that cars are relatively easier to segment compared to the segmentation of other objects. However, the overall segmentation performance remains modest. This is because the CityRefer dataset involves many similar instances placed near each other.

## 5.2 City-level visual grounding

In this experiment, we assess the visual grounding performance, which involves locating the target object within the 3D point clouds based on a given natural language description. To evaluate the performance of both instance segmentation and visual grounding separately, we assume that ground-truth instance segmentation masks are provided. We provide ten candidate answers, including the correct answer, for each description.

**Method.** We modified the InstanceRefer model [56] to enable city-level visual grounding. The modified model follows a four-step process to perform visual grounding. First, we extract object features from each candidate instance by applying a four-layer sparse convolution network [16] with an average pooling layer to input 3D points. Second, we use a one-layer bi-directional GRU (BiGRU) [13] to extract language features from the given description. Third, object features and language features are concatenated and fed into another BiGRU to obtain visual-language features that represent the relationship between the description and the 3D objects. Finally, we compute scores using a two-layer MLP. These scores indicate the likelihood of a candidate instance being the correct grounding for the given description. We use the cross-entropy loss for training. For more comprehensive information regarding the model architecture and training process, please refer to the Appendix.

**Evaluation metrics.** We assess the accuracy of our predictions by comparing their intersection over union (IoU) with the ground truth values. Specifically, we focus on positive predictions that exhibit a higher IoU with the ground truth instances than a certain threshold $k$. We use the Acc@$k$IoU metric, which is commonly used in the field of indoor 3D visual grounding research [9]. For our experiments, we set the threshold value $k$ for IoU to 0.25.

**Results.** Table 6 presents the visual grounding performance for the four target categories, comparing the results of Random (random guess) and Baseline (the method described above). Baseline + Land incorporates landmark features extracted from 3D points and the names of landmark objects, in addition to the object and language features. We see that the baseline methods significantly perform better than the random guess but there is still a large gap between the system performance and human performance (Acc. = 0.950). This demonstrates that city-level visual grounding is a challenging task despite advances in learning technology. Developing large 3D-vision-language models for city-level visual grounding would be a potential future research direction. Figure 8 illustrates qualitative examples of 3D visual grounding when incorporating landmark information.

Table 6: City-level 3D visual grounding performance.

| Method | Building | Car | Ground | Parking | Overall |
|---|---|---|---|---|---|
| Random | $0.103 \pm 0.008$ | $0.103 \pm 0.006$ | $0.091 \pm 0.017$ | $0.094 \pm 0.010$ | $0.101 \pm 0.005$ |
| Baseline | $\mathbf{0.255} \pm 0.005$ | $0.282 \pm 0.010$ | $0.477 \pm 0.024$ | $0.835 \pm 0.034$ | $0.312 \pm 0.006$ |
| Baseline + Land | $\mathbf{0.255} \pm 0.008$ | $\mathbf{0.298} \pm 0.007$ | $\mathbf{0.489} \pm 0.009$ | $\mathbf{0.853} \pm 0.020$ | $\mathbf{0.320} \pm 0.005$ |
| Humans | 0.947 | 0.956 | 0.937 | 0.945 | 0.950 |

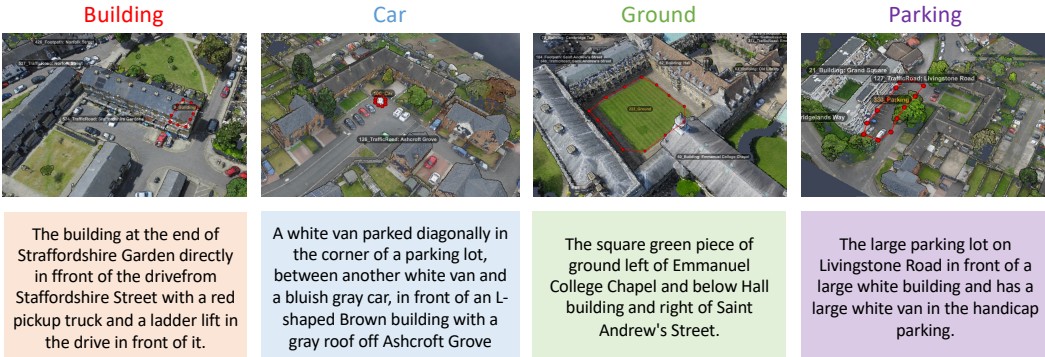

Figure 8: Qualitative examples.

The examples demonstrate how the inclusion of landmark information aids in accurately identifying the target objects. This observation further reinforces incorporating landmark information is a promising way to improve the city-level visual grounding performance.

## 6 Conclusion

We have introduced the CityRefer dataset, a dataset for city-level visual grounding tasks. This dataset offers a comprehensive 3D environment where instance-wise segmentation masks, along with geographic data and labels, are provided. The creation of this environment involved performing a spatial join between the SensatUrban environment and the OpenStreetMap, resulting in a rich and realistic urban setting. We also provided 35k natural language descriptions to locate objects as well as baseline systems for instance segmentation and visual grounding.

**Limitations and future work.** In this work, we tackled visual grounding in the 3D environment of real cities. However, there are still gaps between the real world and the 3D environment. In particular, because all 3D scenes are static, the vocabulary of the CityRefer dataset is limited to words that specify static objects. To achieve more comprehensive and real-world applicability, it would be necessary to extend the dataset to include dynamic environments, where objects and scenes can change over time. Furthermore, while our visual grounding system has demonstrated promising results, there is still room for improvement. The main purpose of this work is to introduce a new dataset and indeed it is nontrivial to apply visual grounding systems for our dataset; however, developing more accurate systems would be worth pursuing in the future. We hope that the CityRefer dataset promotes further research and development as well as discussion on geography-aware learning technologies.

**Broader impacts.** Although the dataset is constructed on the basis of a publicly available 2D map (OpenStreetMap), visual grounding in general may result in privacy issues or racial and gender biases. The natural language descriptions we collect are carefully checked so that they do not include private information or offensive text.

## Acknowledgements

This work was supported by JST PRESTO JPMJPR22P8 and JPMJPR20C2, and JSPS KAKENHI 22K12159.

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
