# CityRefer Datasheet

We follow the guidelines of the datasheets for datasets [1] to explain the composition, collection, recommended use case, and other details of the CityRefer dataset.

## A Motivation

**For what purpose was the dataset created?**
We created this CityRefer dataset to facilitate research toward city-scale 3D visual grounding.

**Who created the dataset (e.g., which team, research group) and on behalf of which entity (e.g., company, institution, organization)?**
This dataset was created by Taiki Miyanishi (ATR), Fumiya Kitamori (Tokyo Institute of Technology), Shuhei Kurita (RIKEN), Jungdae Lee (Tokyo Institute of Technology), Motoaki Kawanabe (ATR), and Nakamasa Inoue (Tokyo Institute of Technology).

**Who funded the creation of the dataset?**
This work was supported by JST PRESTO JPMJPR22P8 and JPMJPR20C2, and JSPS KAKENHI 22K12159.

## B Composition

**What do the instances that comprise the dataset represent?**
CityRefer contains descriptions for 3D visual grounding on large-scale point cloud data. We do not provide the 3D point cloud data, which can be downloaded from the official site of SensatUrban [2].

**How many instances are there in total (of each type, if appropriate)?**
There are 5,866 objects on the 3D map with their instance masks. There are 35,196 natural language descriptions for visual grounding.

**Does the dataset contain all possible instances or is it a sample (not necessarily random) of instances from a larger set?**
Landmark objects are sampled from OpenStreetMap[3]. They are representative of all the possible geographical objects.

**Is there a label or target associated with each instance?**
Yes.

**Is any information missing from individual instances?**
No.

**Are relationships between individual instances made explicit (e.g., users' movie ratings, social network links)?**
Yes. We provide metadata for each object.

**Are there recommended data splits (e.g., training, development/validation, testing)?**
Yes. We provide metadata of data splits.

**Are there any errors, sources of noise, or redundancies in the dataset?**
Please refer to the "Quality control" in Sec. 3.2.

**Is the dataset self-contained, or does it link to or otherwise rely on external resources (e.g., websites, tweets, other datasets)?**
We follow prior work [3] and provide descriptions for 3D visual grounding.

**Does the dataset contain data that might be considered confidential?**
No.

---

[3] https://www.openstreetmap.org

**Does the dataset contain data that, if viewed directly, might be offensive, insulting, threatening, or might otherwise cause anxiety?**
No.

## C Collection Process

The collection procedure, preprocessing, and cleaning are explained in Sec. 3 of our main paper.

**Who was involved in the data collection process (e.g., students, crowdworkers, contractors), and how were they compensated (e.g., how much were crowdworkers paid)?**
Data collection and filtering are done by crowdworkers. Data curation is done by coauthors.

**Over what timeframe was the data collected?**
The data was collected between January 2023 to April 2023.

## D Uses

**Has the dataset been used for any tasks already?**
Yes. We have used the CityRefer database for city-scale 3D visual grounding. Please refer to Sec. 5 in our main paper.

**Is there a repository that links to any or all papers or systems that use the dataset?**
Yes.

**What (other) tasks could the dataset be used for?**
Our dataset is primarily intended to facilitate research in 3D visual grounding. However, it can also be broadly applicable to 3D and language tasks such as 3D object retrieval, 3D question answering, 3D dense captioning, language-guided navigation, embodied question answering, etc.

**Is there anything about the composition of the dataset or the way it was collected and preprocessed/cleaned/labeled that might impact future uses?**
Nothing.

**Are there tasks for which the dataset should not be used?**
It should not be used as a tool to monitor individuals without regard for their privacy.

## E Distribution

**Will the dataset be distributed to third parties outside of the entity (e.g., company, institution, organization) on behalf of which the dataset was created?**
Yes.

**How will the dataset will be distributed (e.g., tarball on website, API, GitHub)?**
The CityRefer dataset and our baseline code can be downloaded from our webpage[4] under CC-BY4.0 license and MIT license, respectively.

**Have any third parties imposed IP-based or other restrictions on the data associated with the instances?**
No.

**Do any export controls or other regulatory restrictions apply to the dataset or to individual instances?**
No.

## F Maintenance

**Who will be supporting/hosting/maintaining the dataset?**
The authors will be supporting, hosting, and maintaining the dataset.

---

[4] https://github.com/ATR-DBI/CityRefer

**How can the owner/curator/manager of the dataset be contacted (e.g., email address)?**
The contact email address can be found on our website.

**Is there an erratum?** No. We will provide the erratum as soon as the need arises.

**Will the dataset be updated (e.g., to correct labeling errors, add new instances, delete instances)?**
Yes.

**If the dataset relates to people, are there applicable limits on the retention of the data associated with the instances (e.g., were the individuals in question told that their data would be retained for a fixed period of time and then deleted)?**
N/A.

**Will older versions of the dataset continue to be supported/hosted/maintained?**
Yes.

**if others want to extend/augment/build on/contribute to the dataset, is there a mechanism for them to do so?** N/A.

## References

[1] T. Gebru, J. Morgenstern, B. Vecchione, J. W. Vaughan, H. Wallach, H. Daume III, and K. Crawford. Datasheets for datasets. *Communications of the ACM*, Volume 64, Issue 12, pp. 86–92, 2021.

[2] Q. Hu, B. Yang, S. Khalid, W. Xiao, N. Trigoni, and A. Markham. Sensaturban: Learning semantics from urban-scale photogrammetric point clouds. *Springer International Journal of Computer Vision (IJCV)*, 130:316–343, 2022.

[3] D. Z. Chen, A. X. Chang, and M. Nießner. Scanrefer: 3d object localization in rgb-d scans using natural language. In *Proceedings of the Springer European Conference on Computer Vision (ECCV)*, 2020.

# CityRefer: Supplementary Material

This is supplementary material for the paper: *CityRefer: Geography-aware 3D Visual Grounding Dataset on City-scale Point Cloud Data*. We present additional details of the dataset, instance segmentation, and 3D visual grounding. We also describe additional ablation studies with qualitative results.

## A  Dataset details

### A.1  Data Collection Interface

We developed an annotation website on the Amazon Mechanical Turk platform for language annotation and manual 3D visual grounding. Figure 11 shows the annotation interface for describing target objects. To collect natural language descriptions about the target object in the 3D map, we ask annotators to describe the target object following the given instructions written in the annotation interface. We also provide examples of geographical objects and their corresponding descriptions. For quality-control purposes, we also ask annotators to check if the specified object: (i) is too tiny to write captions for, (ii) differs from the specified object type in the tag, or (iii) corresponds to multiple objects (e.g., two cars are red-lined). Authors manually confirmed and removed the incorrect data for 5 out of 6 objects checked by annotators.

### A.2  Quality Control Details

To further improve the quality of the annotations, we filter out inappropriate descriptions using a manual 3D visual grounding website and re-annotate them. After collecting the initial descriptions, we present the 3D map, along with the corresponding object names and IDs, to the workers. Figure 12 shows the annotation interface for 3D visual grounding. The workers are instructed to enter the object IDs that best match the provided descriptions for the 3D map. In addition, they are prompted to check a box if no object in the 3D map matches the description or if multiple objects correspond. We discard such incomplete descriptions and re-annotate the corresponding objects using the annotation website used during the initial annotation step, as shown in Figure 11. To ensure comprehensive coverage, we collect six descriptions for each object, thereby capturing multiple perspectives and linguistic variations.

## B  Instance Segmentation Details

### B.1  Architecture

We used the SoftGroup++ architecture [1], an extension of SoftGroup for our instance segmentation task. Figure 9 shows the overview of SoftGroup++. The approach consists of two main stages: bottom-up grouping and top-down refinement. Initially, point features are extracted from the input point clouds using a U-Net backbone. Next, semantic scores and offset vectors are predicted by the semantic and offset branches, respectively. A soft grouping module then uses these predictions to generate instance proposals. The feature extractor layer extracts backbone features from these proposals, which a tiny U-Net subsequently processes. Finally, the classification, segmentation, and mask-scoring branches are used to derive the final instances. For our experiments, we used the official implementation[5] of SoftGroup++ and customized the dataset configuration to suit the SensatUrban dataset.

---

[5] https://github.com/thangvubk/SoftGroup/tree/softgroup++

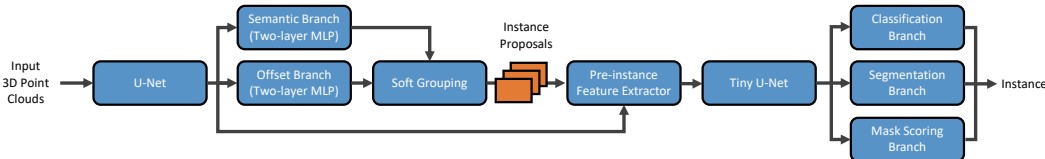

Figure 9: SoftGruop++ architecture [1] for instance segmentation.

## B.2 Training

We conducted training of SoftGroup++ on a large-scale point cloud dataset containing a total of 5,866 objects belonging to the 'Ground,' 'Building,' 'Parking,' and 'Car' categories. To achieve this, we adopted the configuration used, for instance segmentation in STPLS3D [5], a large-scale synthetic 3D point cloud dataset. To ensure computational efficiency, we downsampled the input point clouds uniformly to a ratio of 1/50. Additionally, we cropped the point clouds into non-overlapping blocks, each covering an area of $250m^2$. During training, we initialized the learning rate to 4e-3 and used a cosine annealing scheduler to adjust it. The training process was executed on a single node equipped with four V100 GPUs, using FP16 mixed precision for improved computational performance. All hyperparameters used in our training setup are listed in Table 7.

Table 7: Hyperparameters for training the 3D instance segmentation model.

| Hyperparameter | Value |
|---:|:---|
| Training epoch | 108 |
| Optimizer | Adam [3] |
| Learning rate | 4e-3 |
| Batch size per GPU | 4 |
| Voxel size | 0.33 |
| Number of semantic classes | 4 |

## C  3D Visual Grounding Details

We present details of our geography-aware 3D visual grounding model. All the neural networks in our implementation were developed using PyTorch v1.31. As our baseline method, we relied on the code provided by InstanceRefer [2][6] and made appropriate modifications to suit the city-scale 3D visual grounding task.

### C.1  Architecture

We developed a CityRefer model consisting of language & 3D object encoders along with an object localization module. Figure 10 provides an overview of our model architecture. In formal terms, we define the inputs to our model as follows: language description $D$, landmarks $L$, and candidate objects $O$ in the 3D map.

**Language encoder.** To process the description, we begin by tokenizing it into tokens $\{w_i\}_{i=1}^{n_d}$ using the BertTokenizer[7]. We then perform projection to obtain word representations $W \in \mathcal{R}^{n_d \times 128}$. Here, $n_d$ represents the number of tokens in the description. These representations are subsequently fed into a one-layer bidirectional GRU (BiGRU) for word sequence modeling. We use the first hidden state from the BiGRU as the sentence embedding for the description, denoted as $s \in \mathcal{R}^{1 \times 128}$.

**3D object encoder.** We use the combined 3D data consisting of point coordinates and colors to represent the point cloud of each object on the 3D map. To extract object features from the point cloud of the object candidates, we encode them using SparseConv [4]. We use average pooling to

---

[6] https://github.com/CurryYuan/InstanceRefer
[7] https://huggingface.co/docs/transformers/model_doc/bert

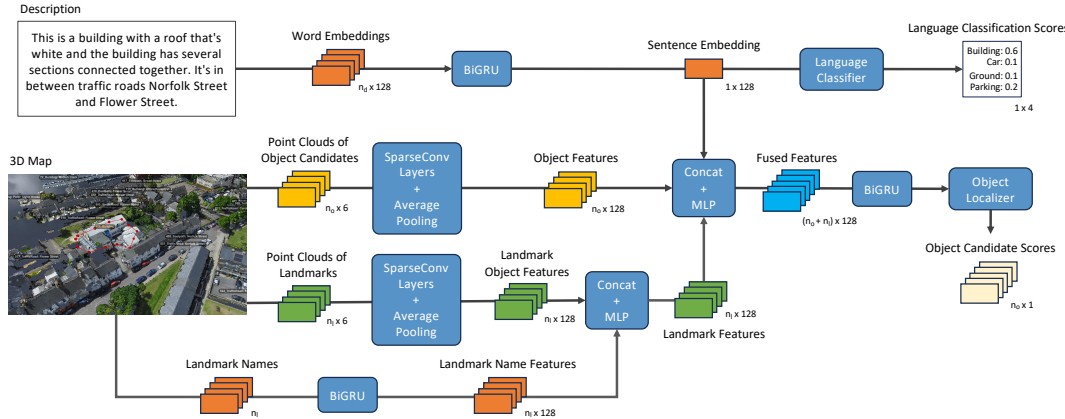

Figure 10: CityRefer architecture for 3D visual grounding.

obtain object features denoted as $O \in \mathcal{R}^{n_o \times 128}$, where $n_o$ represents the number of object candidates targeted for visual grounding. Similarly, we encode landmark objects using SparseConv, resulting in landmark features $L_o \in \mathcal{R}^{n_l \times 128}$. Here, $n_l$ denotes the number of landmarks present in the 3D map. Furthermore, we tokenize and encode the landmark names (e.g., 'Clare College Conferencing') using a BiGRU. We obtain the sentence embeddings of the landmark names as landmark name features, denoted as $L_n \in \mathcal{R}^{n_l \times 128}$. To incorporate both landmark object and name features, we concatenate and fuse them using a multi-layer perceptron. This fusion process yields the final landmark features $L \in \mathcal{R}^{n_l \times 128}$.

**Object localization module.** The fused features, combining the object and landmark features, are further combined with the sentence embeddings of the input descriptions. The resulting fused features are then fed into a BiGRU to establish associations between the object and landmark features. Subsequently, a softmax function is applied to the output of the BiGRU within the object localization module. This step generates scores for the object candidates. As a result, the CityRefer model outputs the object candidate with the highest score as the target corresponding to the given description.

## C.2 Training

To train our 3D visual grounding model, we use a dataset consisting of 35k natural language descriptions of 3D objects. During training, we use the Adam optimizer [3] with a learning rate of 1e-4. The training process is executed on a single V100 GPU, using FP16 mixed precision through the PyTorch native amp module. All hyperparameters are summarized in Table 8.

Table 8: Hyperparameters for training 3D visual grounding model.

| Hyperparameter | Value |
| --- | --- |
| Training epochs | 30 |
| Optimizer | Adam [3] |
| Learning rate | 1e-4 |
| Weight decay | 0.5 |
| Batch size per GPU | 64 |
| Number of object candidates | 10 |
| Number of instance points | 1024 |
| Hidden size | 128 |
| Dropout probability | 0.1 |
| Tokenizer | BertTokenizer |
| Point cloud encoder | Sparse Convolutional Networks [4] |

Table 9: Ablation study of the proposed baseline method with different features.

| Method | Building | Car | Ground | Parking | Overall |
|---|---|---|---|---|---|
| Ours (point=512) | $0.252 \pm 0.002$ | $0.288 \pm 0.006$ | $0.469 \pm 0.018$ | $\mathbf{0.879} \pm 0.014$ | $0.314 \pm 0.002$ |
| Ours (point=2048) | $0.229 \pm 0.003$ | $0.291 \pm 0.006$ | $0.473 \pm 0.016$ | $0.846 \pm 0.023$ | $0.302 \pm 0.004$ |
| Ours (wo/ color) | $0.235 \pm 0.006$ | $0.287 \pm 0.007$ | $0.466 \pm 0.013$ | $0.845 \pm 0.033$ | $0.303 \pm 0.004$ |
| Ours (wo/ name) | $0.250 \pm 0.007$ | $0.283 \pm 0.009$ | $0.475 \pm 0.005$ | $0.836 \pm 0.020$ | $0.310 \pm 0.005$ |
| Ours | $\mathbf{0.255} \pm 0.008$ | $\mathbf{0.298} \pm 0.007$ | $\mathbf{0.489} \pm 0.009$ | $0.853 \pm 0.020$ | $\mathbf{0.320} \pm 0.005$ |

## D  Additional Quantitative Analysis

We describe ablation studies conducted on the CityRefer model. Table 9 shows the results of the ablation study using our baseline method (Baseline + Land) with different features.

**Effect of instance size:** We compared our baseline method (Ours), which uses 1024 points, with variants trained using 512 and 2048 points (Ours point=512, 2048). The results indicate that the choice of the number of points in an instance affects the performance of the 3D visual grounding model.

**Effect of point colors:** In our evaluation, we compared the performance of our baseline method (Ours) with a variant trained without RGB values (Ours wo/ color). The results, as shown in the table, clearly demonstrate the effectiveness of color information in city-level 3D visual grounding. The use of RGB values is beneficial, particularly when distinguishing similar objects, such as cars or buildings, where the color of the roof plays a significant role in differentiation.

**Effect of landmark name:** We conducted a comparison between our baseline method (Ours) and a variant trained without landmark names (Ours wo/ name). The results reveal the crucial role played by landmark names in enhancing the accuracy of 3D visual grounding.

## E  Additional Qualitative Analysis

We demonstrate how our 3D visual grounding model works by visualizing examples. Figure 13 shows several typical examples. The results highlight the accurate prediction of the target object based on the provided descriptions, showing the discriminative ability of our model in the context of city-scale 3D visual grounding, thanks to the use of landmark features. For example, in the second column of the first row, even in the presence of multiple white cars within the 3D data, our model effectively can use the geographic information of the road, 'Graham Warren Way,' to narrow down the location of the target object while a method without landmark information fails to make the correct prediction.

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

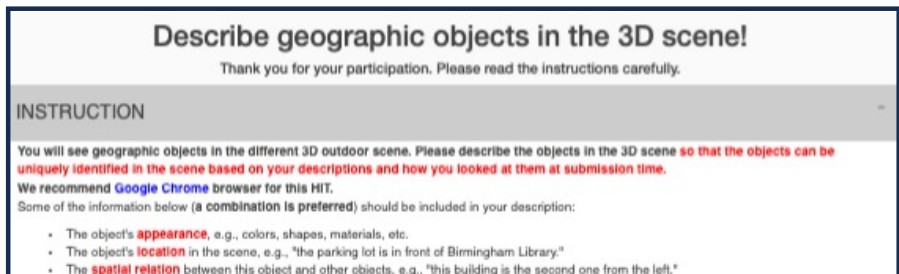

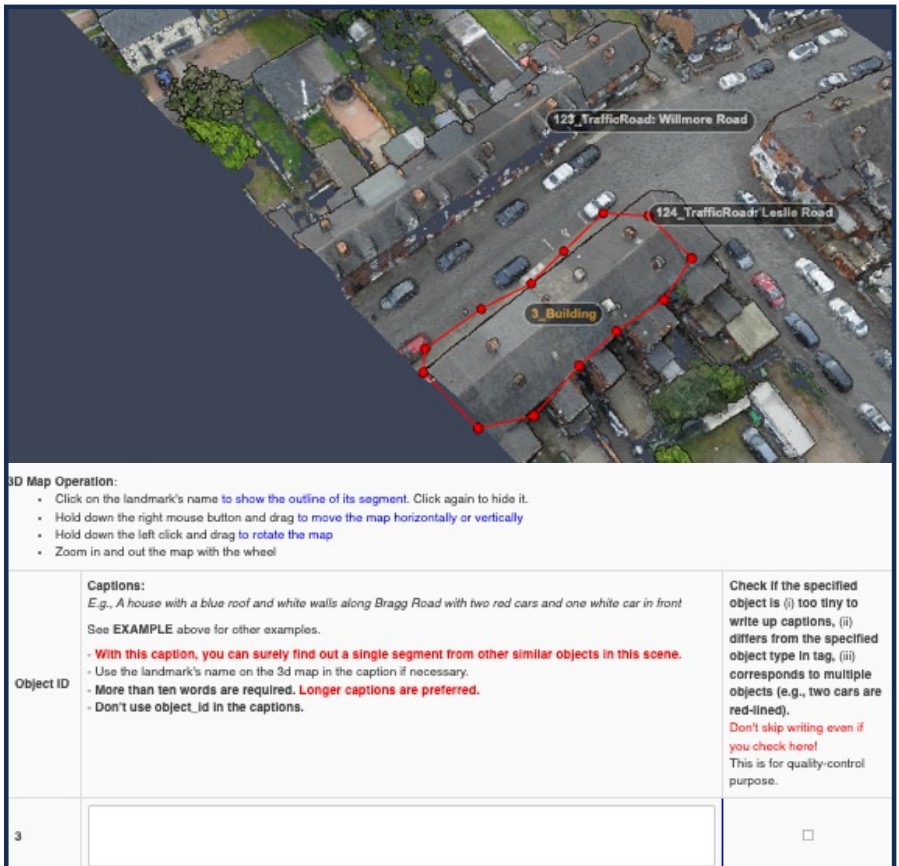

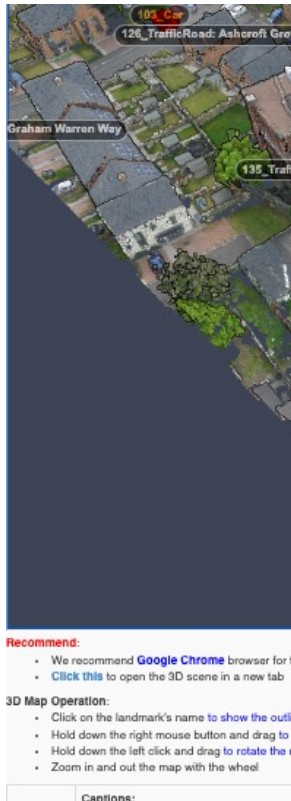

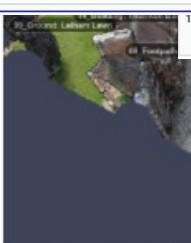

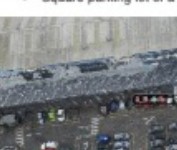

Figure 11: Annotation interface for collecting object descriptions. The top shows an instruction and annotation example part, and the bottom shows the part where workers input descriptions corresponding to given object IDs.

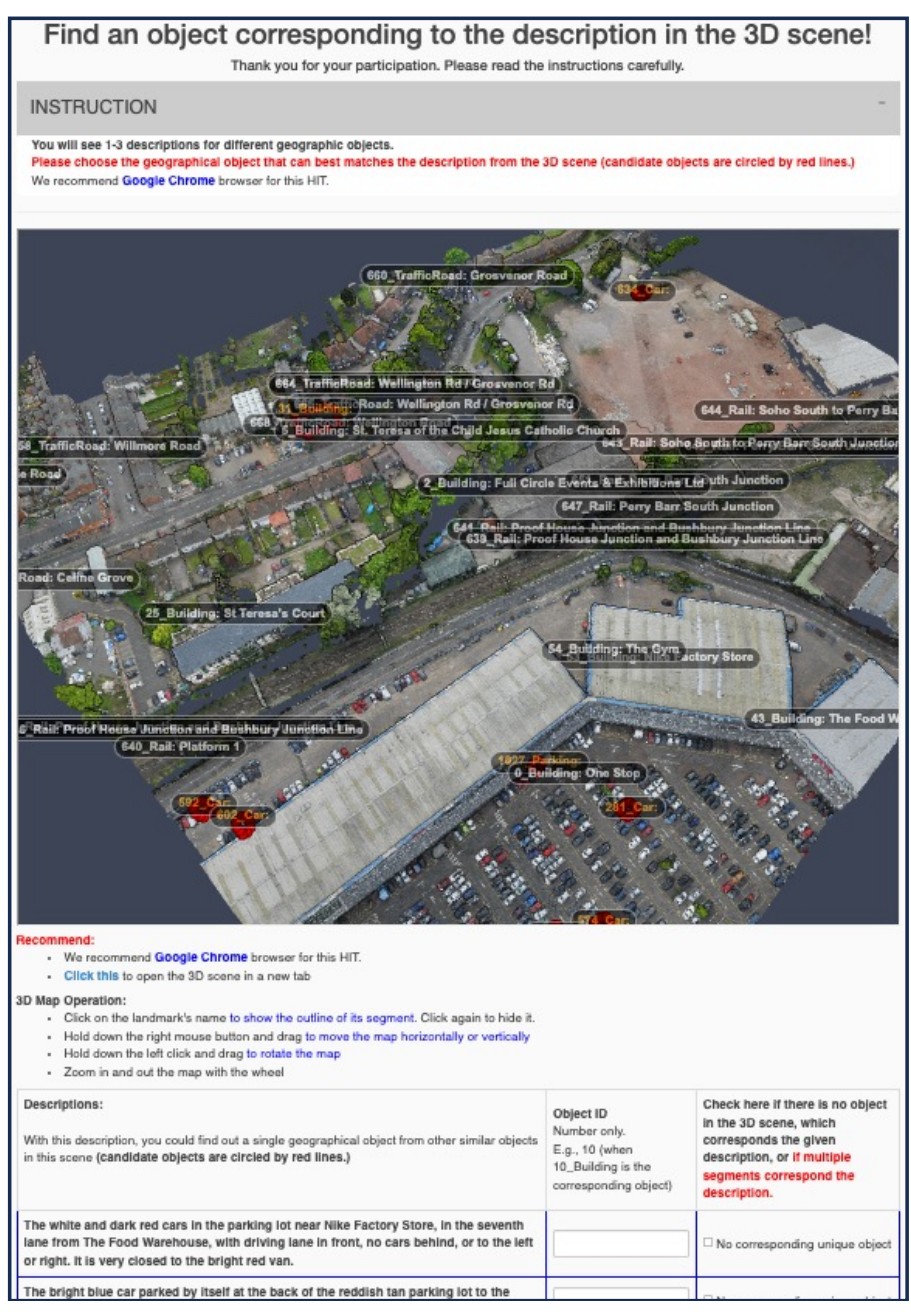

Figure 12: Annotation interface for checking object's descriptions to perform manual 3D visual grounding. Workers input object ID in the 3D map, corresponding to given descriptions.

| Building | Car | Ground | Parking |
|----------|-----|--------|---------|

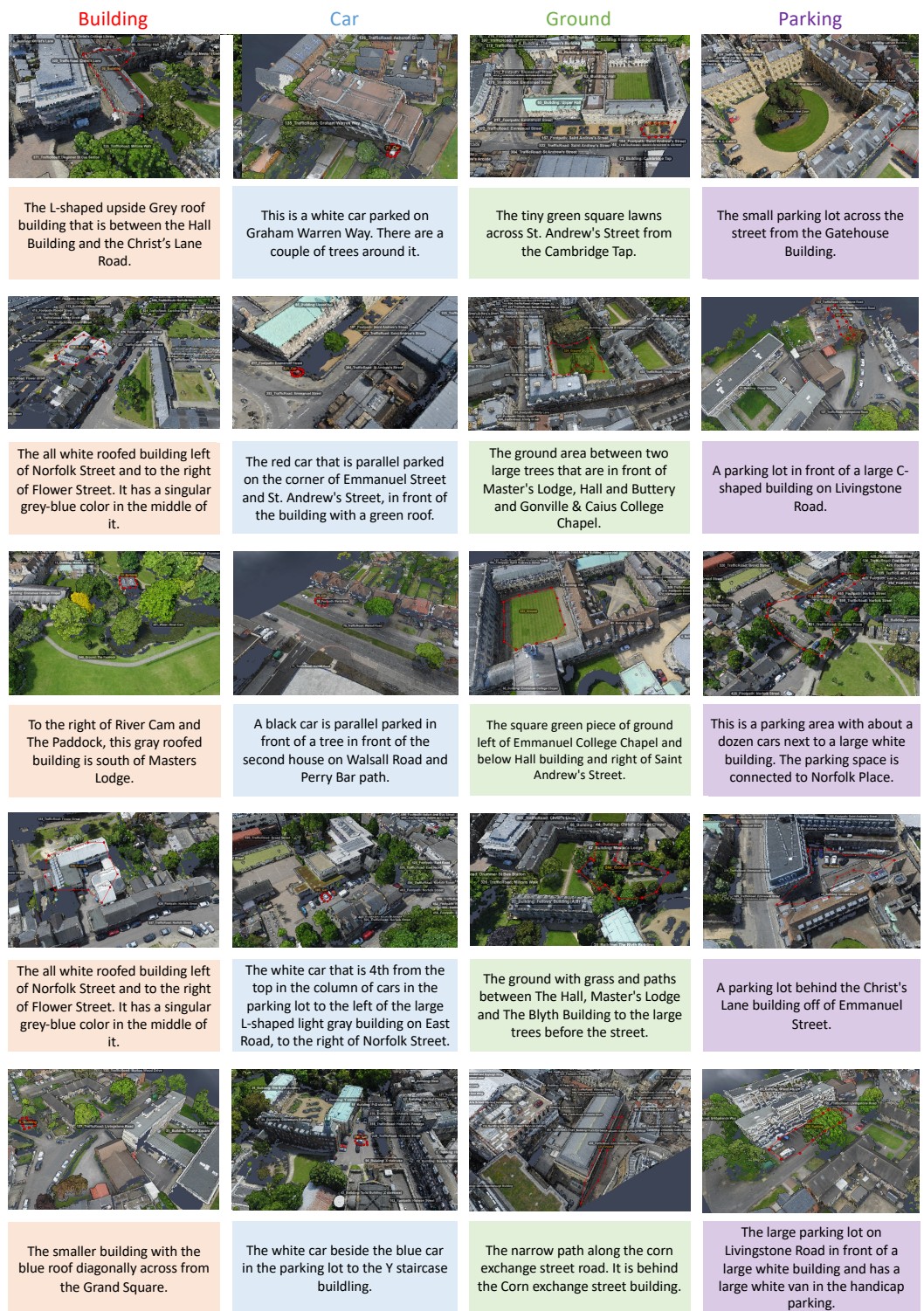

The L-shaped upside Grey roof building that is between the Hall Building and the Christ's Lane Road.

This is a white car parked on Graham Warren Way. There are a couple of trees around it.

The tiny green square lawns across St. Andrew's Street from the Cambridge Tap.

The small parking lot across the street from the Gatehouse Building.

The all white roofed building left of Norfolk Street and to the right of Flower Street. It has a singular grey-blue color in the middle of it.

The red car that is parallel parked on the corner of Emmanuel Street and St. Andrew's Street, in front of the building with a green roof.

The ground area between two large trees that are in front of Master's Lodge, Hall and Buttery and Gonville & Caius College Chapel.

A parking lot in front of a large C-shaped building on Livingstone Road.

To the right of River Cam and The Paddock, this gray roofed building is south of Masters Lodge.

A black car is parallel parked in front of a tree in front of the second house on Walsall Road and Perry Bar path.

The square green piece of ground left of Emmanuel College Chapel and below Hall building and right of Saint Andrew's Street.

This is a parking area with about a dozen cars next to a large white building. The parking space is connected to Norfolk Place.

The all white roofed building left of Norfolk Street and to the right of Flower Street. It has a singular grey-blue color in the middle of it.

The white car that is 4th from the top in the column of cars in the parking lot to the left of the large L-shaped light gray building on East Road, to the right of Norfolk Street.

The ground with grass and paths between The Hall, Master's Lodge and The Blyth Building to the large trees before the street.

A parking lot behind the Christ's Lane building off of Emmanuel Street.

The smaller building with the blue roof diagonally across from the Grand Square.

The white car beside the blue car in the parking lot to the Y staircase buildling.

The narrow path along the corn exchange street road. It is behind the Corn exchange street building.

The large parking lot on Livingstone Road in front of a large white building and has a large white van in the handicap parking.

Figure 13: Additional qualitative examples.