# OpenReview forum: "CityRefer: Geography-aware 3D Visual Grounding Dataset on  City-scale Point Cloud Data"
_NeurIPS.cc/2023/Track/Datasets_and_Benchmarks — NeurIPS 2023 Datasets and Benchmarks Poster_

### Official Review · Reviewer_eXYx · 2023-07-04
**Newly collected data for visual grounding of large point clouds**

**Rating:** 7
**Confidence:** 2
**Clarity:** I found the paper to be very clear.

**Strengths:**

* The paper does a nice job of positioning the gathered data relative to existing work, e.g. using summary statistics.
* The data seems interesting and the data collection is definitely a significant task, i.e. others would probably not collect this data as a side-project for a methods paper.
* The manual verification of the data is helpful.

**Additional Feedback:**

While reading the paper I was wondering if annotators were familiar with the geography of the cities they annotate. I understand that they are most likely not, e.g., local students due to being hired through mechanical turks. I was unable to make up my mind if I thought that having local knowledge would be a pro or a con, and I think it would be nice to have a discussion of such in the paper (local knowledge will influence the text descriptions).

**Correctness:**

I did not see any glaring issues with the dataset collection. I found the instructions given to the MTurkers to be clear and written such as to promote long explicit descriptions.

**Documentation:**

I found the website to be clear and the source code to be reasonably structured. I did not attempt to actually run the code, though.

I did not find a maintenance plan for the data. Here I found it potentially troublesome that the data is hosted on Google Drive, which can be rather brittle (e.g. the hosting depends on the maintainer not running out of space on Google Drive). Here I would have preferred a more proper data host, e.g. a university library, but I acknowledge that such may not always be available.

**Ethics:**

The paper discusses potential privacy issues, and the authors have manually verified the data to avoid such issues. This is imperfect, but I don't think we can ask for more.

**Limitations:**

The paper does a nice job of discussing the limitation that the data is static (unlike actual cities).

**Opportunities For Improvement:**

* The paper introduces train/validation/test splits and provides proposed measures of accuracy. I could see value in providing a curated leaderboard to track results online. This would help future authors determine which is the baseline to beat.
* In order to run the code, it seems I need to manually download the data. It would be helpful if the data loader could automatically download data if it is not already available (to reduce the barrier of entry).

**Relation To Prior Work:**

Yes, existing work is clearly discussed.

The data expands SensatUrban.

**Summary And Contributions:**

The paper introduces the CityRefer data set, which consists of manual annotations and text descriptions of objects in city-scale point clouds. The point clouds were previously available (SensatUrban), so the main contribution is in the human annotations. These annotations include:
* Manual alignment of the point clouds with OpenStreetMap (OSM), thereby providing access to information available in OSM (e.g. street names).
* Text descriptions of objects extracted from segmentation masks.

All data is collected using Mechanical Turks and is manually verified post hoc.

---

> ### Author Response · Authors · 2023-08-22
> **Response**
>
> > The paper introduces train/validation/test splits and provides proposed measures of accuracy. I could see value in providing a curated leaderboard to track results online. This would help future authors determine which is the baseline to beat.
>
> Thank you so much for your positive review and suggestion, we will hide the test split and prepare a leaderboard with EvalAI, etc.
>
>
> > In order to run the code, it seems I need to manually download the data. It would be helpful if the data loader could automatically download data if it is not already available (to reduce the barrier of entry).
>
> Thank you for the good advice. We've added a data download script for user convenience and updated our codebase.
>
> > I did not find a maintenance plan for the data. Here I found it potentially troublesome that the data is hosted on Google Drive, which can be rather brittle (e.g. the hosting depends on the maintainer not running out of space on Google Drive). Here I would have preferred a more proper data host, e.g. a university library, but I acknowledge that such may not always be available.
>
> The CityRefer dataset mainly consists of language annotations and is relatively compact (approximately 10MB). To ensure accessibility, we will update the link on the GitHub repository and provide a data download script that retains a link to the dataset's storage location. Thank you for your suggestions.
>
>
> > While reading the paper I was wondering if annotators were familiar with the geography of the cities they annotate. I understand that they are most likely not, e.g., local students due to being hired through mechanical turks. I was unable to make up my mind if I thought that having local knowledge would be a pro or a con, and I think it would be nice to have a discussion of such in the paper (local knowledge will influence the text descriptions).
>
> Thanks for the interesting feedback. In this work, we did not use geographic information not listed in OpenStreetMap because the main focus of this work was 3D visual grounding based on 3D map and their appearance. The advantage of hiring individuals familiar with local information as annotators is that we can use local information not listed on the map. However, it is difficult to hire such a person in AMTurk. We agree that using such local area information is an interesting approach for future work. We think using Google's local review could be an alternative way to do so.

---

> > ### Comment · Reviewer_eXYx · 2023-08-22
> > **Thanks**
> >
> > Thanks for the follow. I will retain my positive score as I think the paper is a valuable contribution to the community.

---

### Official Review · Reviewer_vMMJ · 2023-07-13
**This paper provides a useful dataset, namely CityRefer, for city-level 3D visual grounding task**

**Rating:** 7
**Confidence:** 4
**Clarity:** The paper is well written.

**Strengths:**

- This paper proposes a new 3D visual grounding dataset for city-level tasks. The dataset first utilizes SensatUrban [19] and OpenStreetMap to perform the spatial join. Therefore, the annotation of building and street names can be directly mapped on the 3D dataset.

- The authors refine the semantic masks in SensatUrban to instance-level masks. They divide semantic categories into three groups, using filtering, clustering (DBSCAN), and detection (YOLOv7) to process different groups separately.

- To annotate the language description for target objects, the authors design an interface for annotation and verification. The quality control can effectively remove incorrect descriptions for verification.

**Additional Feedback:**

Nil.

**Correctness:**

The claims are correct. The dataset is constructed in a sound way and is easy to process with the provided tools and instructions.

**Documentation:**

The authors provide details on data collection and organization. They also proved URLs for datasets, documents, and instructions.

**Ethics:**

No, there are no or only very minor ethics concerns.

**Limitations:**

The authors have addressed the limitations and broader impacts in Sec. 6.

**Opportunities For Improvement:**

The authors run the baseline method InstanceRefer for city-level 3D visual grounding. Actually, current 3D visual grounding methods are all specially designed for indoor datasets. I would like to see some discussions about the challenges (not only results like AP) after directly transferring these methods to large-scale scenes, e.g. computation limitations, training and inference time and efficiency, etc.

**Relation To Prior Work:**

This paper clearly discusses the difference from previous indoor 3D visual grounding datasets.

**Summary And Contributions:**

This paper proposes a dataset for the city-scale 3D visual grounding task.  The dataset provides instance-wise segmentation masks for 5k objects including 1.8k landmark objects with their name labels and 35k natural language descriptions for city-level visual grounding. This paper also uses a baseline to perform city-level 3D visual grounding.

---

> ### Author Response · Authors · 2023-08-22
> **Response**
>
> > The authors run the baseline method InstanceRefer for city-level 3D visual grounding. Actually, current 3D visual grounding methods are all specially designed for indoor datasets. I would like to see some discussions about the challenges (not only results like AP) after directly transferring these methods to large-scale scenes, e.g. computation limitations, training and inference time and efficiency, etc.
>
> Thank you for the good suggestion. Outdoor 3D datasets contain a larger number of objects compared to indoor datasets. Our approach will face limitations related to memory usage and computational time when extending the scope of 3D visual grounding. To remedy this, leveraging landmark information from the description to narrow down the area on a 2D MAP will be effective. Subsequently, we can leverage detailed 3D information to find the target object. We would like to include this discussion in the limitations section.

---

> > ### Comment · Reviewer_vMMJ · 2023-08-29
> > **Thanks for your reply**
> >
> > Thanks for your reply. I will keep my rating.

---

### Official Review · Reviewer_hsUT · 2023-07-20
**CityRefer: Geography-aware 3D Visual Grounding Dataset on City-scale Point Cloud Data**

**Rating:** 7
**Confidence:** 2
**Correctness:** Yes
**Clarity:** Yes

**Strengths:**

* The introduction of the CityRefer dataset addresses a clear gap in city-level 3D visual grounding, making it a significant contribution to the field.
* The ability to linguistically comprehend city scenes has broad applications, particularly in emerging areas like autonomous driving and aerial drone navigation.
* The commitment to release code, data, and trained models enhances the paper's relevance by providing tangible resources for others to build upon.
* It's interest to align a city-scale point cloud with openStreetMap and uses annotations from OSM.


**Additional Feedback:**

None

**Documentation:**

Yes

**Ethics:**

Privacy is a concern for city-scale dataset, but it's more like a problem of SensatUrban

**Limitations:**

Creating city-level datasets might inadvertently capture sensitive or private information. Although the authors have manually filtered the dataset, the process's thoroughness in addressing privacy is not detailed.

**Opportunities For Improvement:**

* While the CityRefer dataset addresses city-level 3D visual grounding, it might be specialized for certain urban environments, limiting its generalizability to varied global cities with different architectural and cultural nuances.
* Little evidence and rationality is provided regarding the techniques and models used in model construction. I would encourage the authors to explain the reasons or add more experiment to prove the choices of techniques.

**Relation To Prior Work:**

I think Table 1 is sufficient

**Summary And Contributions:**

The paper introduces advancements in urban 3D scanning technologies, focusing on the gap in city-level linguistic comprehension of 3D scenes. While large-scale 3D datasets are now common, deriving linguistic representations from them remains challenging. Central to the paper is the "CityRefer dataset" developed for city-level 3D visual grounding. This dataset encompasses 35k natural language descriptions tied to 3D objects in the SensatUrban environment, and 5k object labels, which include landmarks. These advancements help in tasks like language-guided autonomous driving. To populate the dataset, crowdsourcing was utilized, followed by rigorous manual filtering to assure quality. Notably, the dataset offers segmentation masks for objects, integrated using a semi-automatic system linked with OpenStreetMap. Moreover, a baseline system is presented, incorporating geographical data to pinpoint objects effectively. This work is useful in enhancing interactive applications that rely on accurate 3D visual and linguistic grounding at a city scale.

---

> ### Author Response · Authors · 2023-08-22
> **Response**
>
> > While the CityRefer dataset addresses city-level 3D visual grounding, it might be specialized for certain urban environments, limiting its generalizability to varied global cities with different architectural and cultural nuances.
>
> Thank you for your positive review and insightful comments. Our proposed CityRefer dataset comprises language annotations for the point cloud data from two cities in England (Cambridge and Birmingham). As the reviewer pointed out, geographical objects show differences across countries. To enhance the generalizability of 3D visual grounding across diverse global cities, it is crucial to create city-scale 3D visual grounding datasets for various countries by annotating additional city-scale 3D datasets [1, 2, 3]. We agree that generalizing 3D visual grounding to varied global cities is an interesting direction for future work. We believe our CityRefer dataset provides a first step for 3D visual grounding on a city scale across different countries.
>
> [1] DALES (https://arxiv.org/abs/2004.11985)
> [2] Campus3d (https://3d.nus.app/)
> [3] PLATEAU (https://www.mlit.go.jp/plateau/)
>
>
> > Little evidence and rationality is provided regarding the techniques and models used in model construction. I would encourage the authors to explain the reasons or add more experiment to prove the choices of techniques.
>
> We thank the reviewer for the good suggestion. Since the 3D masks of the instance are given in our experiment, we adopted a baseline method inspired by the well-established instance-based 3D visual grounding method that uses 3D instance segments as inputs. Our method can achieve rapid learning and inference while maintaining accuracy for city-scale data. We also conducted preliminary experiments with the other standard 3D object localization methods, but the outcomes were unsatisfactory. To explain the reasons for the choice of technics, we would like to add the details and advantages of the proposed method in Section 5.2 and the supplemental material.
>
>
> > Creating city-level datasets might inadvertently capture sensitive or private information. Although the authors have manually filtered the dataset, the process's thoroughness in addressing privacy is not detailed.
>
> To construct the CityRefer dataset, we used publicly available geographic data from OpenStreetMap. Due to the point cloud's resolution limitations, individuals were either almost non-existent or not identifiable in the 3D data from the SensatUrban dataset.  For the annotation data, we instruct annotators to create descriptions based on geographic information and the appearance of 3D data. When manually checking annotations data, we could not find sensitive or private information. If we receive reports of sensitive or private information on the CityRefer dataset, we plan to remove them and update CityRefer's annotation data.

---

### Official Review · Reviewer_hP2w · 2023-07-23
**Review of CityRefer**

**Rating:** 6
**Confidence:** 4
**Clarity:** The paper is well written and easy to…

**Strengths:**

1. CityRefer distinguishes itself from other outdoor datasets through its larger size, more diverse cityscape scenes, and the inclusion of natural language descriptions.
2. The process of constructing the dataset is systematically organized, incorporating rigorous quality control measures.n the first phase of the data construction pipeline, the result is 2D semantic masks derived from a top-down view, rather than 3D masks, as can be observed in the provided figures. Additionally, the authors are capable of providing 3D bounding box labels, making the dataset applicable to fields like autonomous driving.



**Additional Feedback:**

Given that large-scale 3D visual grounding is more resource-intensive compared to visual grounding using remote sensing images [1, 2, 3], and is not well-suited for real-time application, have the authors drawn any performance comparisons with methods that use top views projected from point clouds?

[1] RRSIS: Referring Remote Sensing Image Segmentation, https://arxiv.org/abs/2306.08625
[2] Yuxi Sun, Shanshan Feng, Xutao Li, Yunming Ye, Jian Kang, and Xu Huang. 2022. Visual Grounding in Remote Sensing Images. In Proceedings of the 30th ACM International Conference on Multimedia (MM '22). Association for Computing Machinery, New York, NY, USA, 404–412. https://doi.org/10.1145/3503161.3548316
[3] Y. Zhan, Z. Xiong and Y. Yuan, "RSVG: Exploring Data and Models for Visual Grounding on Remote Sensing Data," in IEEE Transactions on Geoscience and Remote Sensing, vol. 61, pp. 1-13, 2023, Art no. 5604513, doi: 10.1109/TGRS.2023.3250471.





**Correctness:**

Section 5.2 discusses the visual grounding method proposed by the authors, which utilizes ground-truth object instances as input and generates a confidence score to indicate the likelihood. How is this incorporated into the evaluation metric? Furthermore, could you clarify the definition of Intersection over Union (IoU) in this context?

**Documentation:**

The dataset is documented with a development kit and baseline method codes.

**Limitations:**

1. As the authors wrote, the objects are restricted to static ones and scenes. By incorporating descriptions of dynamic objects like movement direction, speed, and actions, a broader research scope could potentially be opened up.

2. Reasoning on a city-level point cloud requires significant memory consumption, which makes models strike a balance between their range of interest and the sparsity of the point clouds.

**Opportunities For Improvement:**

1. In the first stage of the data construction pipeline, the result is more of 2D semantic masks derived from a top-down view, rather than 3D masks, as can be observed in the provided figures.
2. The authors are encouraged to provide 3D bounding box labels, making the dataset applicable to fields like autonomous driving.



**Relation To Prior Work:**

The authors engage with relevant literature in their work and make a contribution toward outdoor visual grounding tasks.



**Summary And Contributions:**

CityRefer introduces a substantial city-scale dataset geared towards the 3D visual grounding task. The dataset encompasses RGB point clouds of various cities, instance-wise segmentation masks, language descriptions for unique object identification, and optional landmark names. The authors effectively carry out instance segmentation and visual grounding.

The contribution can be summarized as follows:
1. They have developed a comprehensive city-scale dataset for the 3D visual grounding task and therefore address a gap that previously existed with indoor-focused datasets.
2. They provide complete labeling instructions and go through a thorough quality check process.

---

> ### Author Response · Authors · 2023-08-22
> **Response**
>
> > In the first stage of the data construction pipeline, the result is more of 2D semantic masks derived from a top-down view, rather than 3D masks, as can be observed in the provided figures.
>
> We thank the reviewer for pointing this out. The two-dimensional contours in the figure were intended to provide a clear visualization of the object. We agree that the result is more of 2D semantic masks derived from a top-down view. However, we could make 3D masks in the form of three-dimensional shapes of objects, different from just 2D ones. To achieve this, we performed the following steps:  First, we converted the manually assigned 3D semantic labels within the SensatUrban dataset into 2D data on the xy-axis. Subsequently, we spatially integrated these labels with the 2D polygons representing geographical features sourced from OpenStreetMap. The 3D semantic labels corresponding to the overlapping 2D data were then used for generating the 3D instance masks. Finally, we manually removed 3D masks that cannot be identified to ensure the quality of the 3D instances.
>
> We will append a note to the figure caption, clarifying that the 2D contours were generated from a 3D mask for the purpose of visualization. Additionally, we will include instructions on how to create 3D masks.
>
>
> > The authors are encouraged to provide 3D bounding box labels, making the dataset applicable to fields like autonomous driving.
>
> We have added 3D bounding box labels to our updated GitHub page with bounding box visualizations (https://anonymous.4open.science/r/CityRefer-8E13). Thank you very much for your idea.
>
> > Section 5.2 discusses the visual grounding method proposed by the authors, which utilizes ground-truth object instances as input and generates a confidence score to indicate the likelihood. How is this incorporated into the evaluation metric? Furthermore, could you clarify the definition of Intersection over Union (IoU) in this context?
>
> Our method computed confidence scores for N candidate 3D bounding boxes (where we set N=10) and subsequently chose a bounding box with the highest score as the prediction outcome. Following prior research (https://arxiv.org/abs/1912.08830), we calculated the percentage of predicted boxes whose Intersection over Union (IoU) with the ground truth boxes. Here IoU is defined by $\mathrm{IoU}(A, B) = \frac{|A \cap B|}{|A \cup B|}$, $A$ and $B$ are volumes predicted and ground truth 3D bounding boxes.
> Then, we computed accuracy over provided descriptions, considering a prediction as 1 if the IoU threshold exceeded 0.25; otherwise, 0.
>
>
> > Given that large-scale 3D visual grounding is more resource-intensive compared to visual grounding using remote sensing images [1, 2, 3], and is not well-suited for real-time application, have the authors drawn any performance comparisons with methods that use top views projected from point clouds?
>
> Thank you for sharing valuable references. Since the primary objective of our paper is to introduce a task and dataset for large-scale object localization directly in 3D space, we did not conduct a performance comparison with top-view image-based methods. Nevertheless, we also believe that using top-view 2D image data in addition to 3D data is a promising approach for refining a specific region guided by provided linguistic cues. We would like to include suggested papers in the related work section and take them for future method development.

---

> > ### Comment · Reviewer_hP2w · 2023-08-30
> > **Thanks for the response**
> >
> > Thanks for the response. I will keep my rating.

---

### Official Review · Reviewer_vjm6 · 2023-08-01
**Data is valuable, experiments are not enough**

**Rating:** 6
**Confidence:** 3
**Correctness:** Yes
**Clarity:** Yes

**Strengths:**

1. As the text annotations for outdoor scenes are challenging to obtain, they devise a semi-automatic instance mask generation method and manually accompany it with language annotation.
2. They provide a detailed analysis of the data statistics.


**Additional Feedback:**

None.

**Documentation:**

Yes

**Limitations:**

Yes

**Opportunities For Improvement:**

No state-of-the-art methods are evaluated on the proposed dataset. The provided benchmarks are not enough. I suggest the authors to include more recent methods.

**Relation To Prior Work:**

Yes

**Summary And Contributions:**

This paper proposes a city-scale 3D point cloud data for 3D visual grounding.

---

> ### Author Response · Authors · 2023-08-22
> **Response**
>
> > No state-of-the-art methods are evaluated on the proposed dataset. The provided benchmarks are not enough. I suggest the authors to include more recent methods.
>
> Thank you so much for your positive evaluation. We have added a training script and model of a recent method, 3DVG-Transformer+, to our code repository accessible at (https://anonymous.4open.science/r/CityRefer-8E13/models/transformer.py). We chose this method because it has shown high performance in indoor 3D object localization using only 3D data and thus is suitable to evaluate the 3D visual grounding performance only. We implemented this method based on D3Net (https://github.com/daveredrum/D3Net), but we found its performance to be relatively lower than our proposed baseline.

---

### Decision · Program_Chairs · 2023-09-22

**Decision:**

Accept (Poster)

**Comment:**

Five experts reviewed this paper with all accepted recommendations. The area chairs agree that this work makes a very important contribution by introducing a tool for 3D Captioning. The reviewers did raise some valuable concerns that should be addressed in the final camera-ready version of the paper.